# Paediatric Thyroid Carcinoma: The Genetic Revolution and Its Implications for Therapy and Outcomes

**DOI:** 10.3390/cancers17091549

**Published:** 2025-05-02

**Authors:** Joel A. Vanderniet, Noemi A. Fuentes-Bolanos, Yoon Hi Cho, Katherine M. Tucker, Antoinette Anazodo, Andrew J. Bauer, Paul Z. Benitez-Aguirre

**Affiliations:** 1Sydney Medical School, Faculty of Medicine and Health, The University of Sydney, Sydney, NSW 2050, Australiapaul.benitezaguirre@health.nsw.gov.au (P.Z.B.-A.); 2Institute of Endocrinology and Diabetes, The Children’s Hospital at Westmead, Sydney, NSW 2145, Australia; 3School of Clinical Medicine, UNSW Medicine and Health, University of New South Wales, Sydney, NSW 2033, Australia; 4Kids Cancer Centre, Sydney Children’s Hospital, Randwick, NSW 2031, Australia; 5Children’s Cancer Institute, Lowy Cancer Centre, University of New South Wales, Sydney, NSW 2033, Australia; 6Hereditary Cancer Clinic, Prince of Wales Hospital, Sydney, NSW 2031, Australia; 7Nelune Cancer Centre, Prince of Wales Hospital, Sydney, NSW 2031, Australia; 8Division of Endocrinology and Diabetes, The Children’s Hospital of Philadelphia, Philadelphia, PA 19104, USA; bauera@chop.edu; 9Department of Pediatrics, The Perelman School of Medicine, The University of Pennsylvania, Philadelphia, PA 19104, USA

**Keywords:** thyroid nodule risk-stratification, cancer predisposition syndrome, molecularly targeted therapy

## Abstract

Most thyroid carcinomas have genetic changes that are linked to their clinical presentation and behaviour, most notably how likely they are to metastasise and how they may respond to therapy. The prevalence of the genetic changes is different in children and adolescents compared to adults. Over the last decade, the understanding of how these genetic changes are associated with a carcinoma’s ultrasound characteristics, cytology, and histology has improved. This article describes the results of this research and proposes how genetic information can be used to guide the management of thyroid nodules and carcinomas in children and adolescents. The goal of this contemporary, precision-medicine approach is to reduce the extent of surgery and the risk of complications when possible, as well as to optimise the treatment of advanced or treatment-resistant carcinomas through the incorporation of oncogene-specific inhibitory therapy.

## 1. Introduction

Molecular genetics are transforming our understanding of health and disease, as well as disease pathophysiology and management. Our insights into thyroid disease, and particularly thyroid carcinoma, through genetic studies are creating opportunities for personalised medicine through targeted therapies, transforming models of care and improving patient experience and outcomes.

Thyroid nodules are estimated to occur in 1.0–1.5% of children, far less commonly than in adults, and the incidence increases with age [1]. However, the rate of malignancy in paediatric nodules is higher, ~20–25% compared with ~5–10% in adults [2,3], and children are more likely to have lymph node and pulmonary metastasis at diagnosis. There is, therefore, a lower threshold for the biopsy and surgery of paediatric nodules. Most nodules occur in the absence of risk factors, but radiation, chemotherapy, autoimmune thyroiditis, and a genetic predisposition all increase the risk.

At least 95% of thyroid carcinomas arise from follicular cells and are termed differentiated thyroid carcinomas (DTCs), while ~5% arise from the parafollicular C-cells, termed medullary thyroid carcinomas (MTCs). In paediatric DTC, 90% are papillary thyroid carcinomas (PTCs) and the remainder are follicular thyroid carcinomas (FTCs) [4]. Poorly differentiated thyroid carcinomas (PDTCs) and anaplastic thyroid carcinomas are also of follicular origin but rare in paediatrics.

Most paediatric DTCs are sporadic and associated with de novo somatic gene fusions and point mutations that lead to tumorigenesis and progression [5]. The identification of these gene alterations can be informative regarding the risk of malignancy, tumour characteristics, and optimal treatment, including the use of molecularly targeted therapies. However, due to a lack of data, current paediatric guidelines do not include routine comprehensive molecular testing at the time of diagnosis of a thyroid carcinoma [2,6,7].

A minority of DTCs are associated with germline pathogenic or likely pathogenic variants (P/LPV) in cancer predisposition genes (CPGs), and the identification of these may inform clinicians about tumour behaviour, surveillance for other tumours, and risks to family members. Genetic testing in these individuals is based on the assessment of syndromic clinical features and family history, in addition to histopathological features.

In contrast to DTCs, MTCs arise from the parafollicular C-cells, which do not produce thyroglobulin, are not responsive to TSH, and do not express the sodium–iodide symporter, so are not responsive to radioactive iodine (RAI) therapy. They secrete numerous other products, including calcitonin and carcinoembryonic antigens, which are the two main tumour markers used in the diagnosis, response to therapy, and surveillance of MTCs [5,8]. Paediatric MTCs are nearly always associated with multiple endocrine neoplasia type 2 (MEN2), caused by germline *RET* P/LPV. Genetic testing should, therefore, be offered to all children diagnosed with MTCs [5,8]. Sporadic MTCs occur most commonly after age 30 [9].

This review describes the current understanding of genomic alterations implicated in the pathogenesis of paediatric thyroid carcinomas, approaches to molecular testing for paediatric patients with thyroid nodules and carcinomas, evolving treatment strategies unlocked by molecular testing, and the need for further research.

## 2. Somatic Gene Alterations in Differentiated Thyroid Carcinomas

Most paediatric DTCs are associated with de novo gene fusions and point mutations resulting in the constitutive activation of the mitogen-activating protein kinase (MAPK) and phosphoinositide 3-kinase (PI3K) signalling pathways [5]. These gene alterations are mostly somatic, mutually exclusive, and associated with distinct gene expression and clinicopathological characteristics [10]. Paediatric studies have identified gene fusions in 33–65% of carcinomas, especially in younger patients, while they account for only ~15% in adults [11,12,13,14,15]. In contrast, point mutations are very common in adult thyroid carcinomas (~70%), but are found in less than 25% of paediatric carcinomas [16].

In 2013, The Cancer Genome Atlas (TCGA) Research Network published a landmark study, describing the molecular landscape of papillary thyroid carcinomas (PTCs) in nearly 500 adult tumours [10]. Through the multi-omic analysis of thyroid-specific gene expression and the creation of a Thyroid Differentiation Score (TDS), they characterised PTCs as being on a spectrum between *BRAF*-like and *RAS*-like. *BRAF*-like tumours were driven by MAPK-signalling and associated with dedifferentiation, while *RAS*-like tumours displayed concurrent MAPK and PI3K activation and were more well differentiated. This was consistent with clinical observations, such as higher rates of RAI-resistance in *BRAF*-positive tumours and follicular architecture in *RAS*- and *PAX8::PPARG*-positive tumours. More recently, Franco et al. described the need to separately classify *RET* and *NTRK* fusion-driven PTCs in children due to their higher risk of invasion and metastasis compared to *BRAF^V600E^*-driven tumours, despite being neutral to mildly *BRAF*-like tumours on the TDS [14].

### 2.1. RET

*RET* (REarranged during Transfection) gene fusions are more common in children and adolescents than in older adults and are associated with exposure to ionising radiation. The *RET* gene encodes for a tyrosine kinase receptor which is not normally expressed in follicular thyroid cells. The fusion partner activates *RET* in follicular cells, leading to its constitutional activation and tumorigenesis. More than 20 fusion partners have been described, the most common being *PTC* genes [17].

Although *RET* fusions were identified in only 6.3% of PTCs in a TCGA cohort [10], they are the most common molecular alterations in paediatrics, reported in 25–40% of cases across most of the world [12,13,14,15,18,19], although notably 55% in post-Chernobyl studies and only ~10% in post-Fukushima studies [19]. They occur frequently in both sporadic and radiation-induced PTCs of all histological variants, and especially frequently in young patients [12,16,20].

*RET* fusions in adults are associated with a more favourable prognosis and good response to RAI therapy [17]. In children, however, they are associated with invasive disease, including lymph node and distant metastasis, and persistent and recurrent disease [12].

### 2.2. NTRK

The 3 *NTRK* (neurotrophic tyrosine receptor kinase) genes encode for transmembrane receptor proteins that regulate cellular functions through the MAPK, PI3K/AKT, and PLC-gamma pathways. *NTRK* fusions result in the constitutive activation of these pathways, stimulating tumorigenesis and progression [21]. *NTRK* fusions are more common in paediatric thyroid carcinomas than in adult disease and are associated with invasive disease. They are almost exclusively found in PTCs [22].

TCGA identified *NTRK* fusions in only 2.4% of thyroid carcinomas and, at that time, they were said to be more prevalent in radiation-induced carcinomas than sporadic PTCs [10]. A recent study of 144 paediatric thyroid carcinoma patients reported *NTRK* fusions in 13.9% [22]. All in both cohorts were *NTRK1* and *NTRK3* fusions, consistent with prior reports of *NTRK2* fusions predominantly being found in CNS tumours. All the paediatric cases were PTCs: 70% classic variant, 20% widely invasive follicular variant, and 10% diffuse sclerosing variant. Sixty per cent had a non-nodular, diffusely infiltrative appearance on ultrasound. Eighty per cent had cervical lymph node metastasis and 45% pulmonary metastasis, which was associated with an 86% risk of persistent disease one year after the last completed intervention.

Pekova et al. reported *NTRK* fusions in 18% of 93 paediatric PTCs [12]. Interestingly, none of the 14 *NTRK3*-positive tumours had distant metastases and only 8 had lymph node metastases. Nies et al. reported *NTRK* fusions in 26% of 69 paediatric PTCs with distant metastasis. All had persistent structural disease at the time of reporting [23].

### 2.3. ALK

Anaplastic lymphoma kinase (ALK) receptor tyrosine kinase is involved in the regulation of cell proliferation and survival. *ALK* fusions resulting in the constitutive activation of the receptor have been implicated in many non-thyroidal tumours [24]. They have been found in <3% of adult PTCs (0.8% of TCGA [10]) but up to 7% of paediatric PTCs [17,19]. Similar to other fusion oncogenes, they have been associated with more invasive clinicopathological features [12,25], although the available data are limited. *ALK* fusions have also been described in PDTCs and MTCs [26,27].

### 2.4. BRAF

The vast majority of *BRAF* mutations involve a substitution of valine to glutamate at the 600th residue (V600E), resulting in constitutive activation of the MAPK pathway [10]. Although *BRAF* point mutations constitute approximately 60% of all PTC driver mutations, they are much less common in children than adults. *BRAF* has been shown to downregulate genes critical for RAI responsiveness, including thyroid-stimulating hormone receptor (TSHR), sodium–iodide symporter, and thyroglobulin. In adults, *BRAF^V600E^* point mutations are strongly associated with a higher risk of disease progression, recurrence, and RAI-resistance [5,28].

*BRAF^V600E^* mutations are found in only 15–25% of childhood thyroid carcinomas in most parts of the world, except for Asia, where the overall frequency is ~38%, and a rate of 70% was reported in the population studied following the Fukushima nuclear disaster (most of which were PTCs < 2 cm detected by ultrasound screening and likely not radiation-induced) [19]. They are more frequent in sporadic than radiation-induced carcinomas [16], and very few *BRAF* mutations were identified in the population exposed to the Chernobyl nuclear disaster [20,29]. They are particularly rare in children < 10 years of age, and incidence increases with age through adolescence [19,30]. *BRAF^V600E^* mutations in children are associated with a high risk of regional metastasis but a low risk of distant metastasis and RAI-resistance [31,32]. This may be due to the rarity of *TERT*-promotor mutation co-occurrence in paediatric PTCs.

Gene fusions involving *BRAF* also occur infrequently: 2.3% of PTCs in the TCGA study [10] and up to 4% in sporadic paediatric PTCs, although higher rates have been reported in Brazil and in post-Chernobyl studies [17].

### 2.5. TERT

Telomerase reverse transcriptase (*TERT*) maintains the telomere structure at DNA ends but is not expressed in most human somatic cells. C228T and C250T mutations in the *TERT* promoter region (*TERT*-p) occur in numerous cancers and result in the production of the TERT protein, leading to cell immortalisation, a hallmark of cancer cells [33]. They occur in 10–20% of adult DTCs, including 9.4% in TCGA [10], and are associated with increased tumour invasiveness and recurrence, especially when coexisting with *BRAF^V600E^* mutations [17]. *TERT* induces cancer cell dedifferentiation, and a proposed mechanism of action is mediated by enhanced ribosome biogenesis [34]. *TERT*-p mutations are uncommon in paediatric thyroid carcinomas, including in several large paediatric studies that did not identify any *TERT*-p mutations [35,36]. In contrast, a study in China identified C228T mutations in 27% of 48 PTCs in patients aged 3 to 14 years, and they were strongly correlated with tumour invasiveness. However, they found no C250T mutations [37].

### 2.6. RAS

The *RAS* (Rat Sarcoma) family of genes includes *NRAS*, *HRAS*, and *KRAS*. Point mutations in these genes result in the constitutive activation of the MAP3K and PI3K signalling pathways [17]. They are observed commonly in both benign and malignant adult thyroid nodules, including 40–53% of FTCs [38] and 13% of PTCs (mostly follicular variant PTCs [fvPTC]) [10]. They are rare in children, however, accounting for <10% of DTCs [19], again, mostly fvPTCs and FTCs [14]. *RAS* mutations are associated with reduced metastatic behaviour in adults, and it appears this also holds true in paediatrics, although the association is not well described due to low numbers [14].

### 2.7. Pax-8/PPAR-Gamma

Pax-8 is a transcription factor necessary for normal thyroid development and the expression of many thyroid-specific genes, including *TG*, *TPO*, and *SLC5A5*. PPAR-gamma is a nuclear receptor transcription factor that regulates adipogenesis and modulates lipid metabolism and insulin sensitivity [39]. The *PAX8::PPARG* fusion transcript is driven by the *PAX8* promoter, which is highly active in thyroid follicular cells, resulting in high levels of expression of *PAX8::PPARG* mRNA and the Pax-8/PPAR-gamma fusion protein, which acts as an oncoprotein [39].

*PAX8::PPARG* is primarily associated with follicular-pattern thyroid carcinomas, found in approximately one third of adult FTCs and fvPTCs [39], and is associated with more extensive capsular and vascular invasion compared to *RAS*-positive tumours [5]. In children, there are no reports of detecting *PAX8::PPARG* in FTCs; however, it has been detected in 9% of fvPTCs [14,18,40,41].

## 3. Germline P/LPV Associated with Predisposition to Thyroid Carcinomas

Approximately 5% of paediatric thyroid carcinomas are attributed to heritable germline P/LPV conferring an increased risk of a thyroid carcinoma [5]. The Dutch Pediatric Thyroid Cancer Consortium recently reported a 13% prevalence of P/LPV in known CPGs in a cohort of 97 patients with DTCs [42]. However, only 5 of the 13 P/LPV were in genes known to cause DTCs (4 *DICER1* and 1 *APC*). The remainder were in *CHEK2*, *BRCA2*, *HOXB13*, and *MITF*, which are not proven to be causative in DTCs, so may be bystander genes.

The inheritance of a CPS is predominantly autosomal dominant with highly variable penetrance [43]. The age of thyroid carcinoma diagnosis varies widely in most CPSs; diagnosis in childhood is uncommon, except in DICER1 syndrome and PTEN hamartoma tumour syndrome (PHTS).

Familial non-medullary thyroid carcinoma (FNMTC) denotes a pattern of non-syndromic familial DTCs with autosomal dominant inheritance and clinical anticipation, in which subsequent generations present with earlier and more invasive disease [44]. It is defined by the presence of either three first-degree relatives with a follicular-derived thyroid carcinoma, or PTCs in two or more first-degree relatives, where the diagnostic patient is <33 years of age, the affected family members are <45 years of age, there are increased numbers of young males with PTCs in the kindred compared to sporadic PTCs, and the PTCs are multifocal and/or bilateral, with a background combination of benign thyroid nodules [45]. This must be in the absence of a history of ionising radiation or a CPS. No causative single CPG for this entity has been identified.

A description of some important syndromes that predispose to thyroid carcinomas follows.

### 3.1. PTEN Hamartoma Tumour Syndrome

Phosphatase and Tensin Homolog (*PTEN*) is a tumour suppressor gene that regulates cell proliferation via the PI3K-AKT pathway. Somatic variants are found in a wide variety of neoplasms, including 0.5% of PTCs [10], and are estimated to occur in ~6% of human cancers [46]. Germline inactivating variants result in an increased risk of benign and malignant tumours of the breast, uterus, kidney, intestine, skin, thyroid, and others [47]. A number of related clinical syndromes have been associated with *PTEN* germline P/LPV, including Cowden syndrome, Bannayan–Riley–Ruvalcaba syndrome, and Proteus-like syndrome. These syndromes are associated with a wide phenotypic variation, increased tumour risk, macrocephaly, vascular malformations, and neurological manifestations, such as autism, learning difficulties, and seizures [47].

A lifetime risk for benign thyroid conditions of up to 75% has been reported [46]. The cumulative risk of DTCs is 3–5% by age 20, rising to 21–38% by age 70, with a median age of diagnosis of between 31 and 37 years, although diagnoses as young as 7 years have been reported [48]. A high proportion of FTCs and fvPTCs is described. A retrospective natural history study found a 44% prevalence of thyroid nodules ≥10 mm in 50 patients <19 years with a genetic diagnosis of PHTS, two of which were low grade carcinomas [47]. Nodules were rare at <7 years of age, and the prevalence increased with age, particularly during the pubertal years. The progression of nodules was generally slow—of 14 patients with a 5–10 mm thyroid nodule, 57% of the nodules grew to ≥10 mm in a median of 2 years (three within 1 year), while of 28 patients with no thyroid nodules ≥5 mm, 25% developed a nodule ≥10 mm after a median of 5.6 years (none within 2 years).

Diagnostic criteria for PHTS have been published by the National Comprehensive Cancer Network, using various combinations of major and minor criteria [49]. The Cleveland Clinic published a paediatric score to select patients for *PTEN* testing based on prospective data [50]. As an example, based on their calculator, a thyroid carcinoma in combination with macrocephaly is sufficient to warrant genetic testing.

Thyroid surveillance recommendations for individuals with PHTS have varied widely due to the rare occurrence of regional or distant metastasis, as well as the concern regarding thyroid surgery that does not appear to improve the disease-free survival while placing patients at risk for potential complications [51]. The European Reference Network for Genetic Tumour Risk Syndromes, in 2020, recommended commencing annual thyroid ultrasounds at the age of 18 years [52]. Two recently published guidelines recommend annual physical examinations and the commencement of ultrasound surveillance at age 12 years, repeating every 3 years in the absence of nodules. If nodules are detected, age-appropriate guidelines for evaluation and management should be followed [53,54].

### 3.2. DICER1 Syndrome

DICER1 syndrome is caused by heterozygous germline P/LPV in *DICER1*. In most DICER1-related tumours, there is a heterozygous germline loss-of-function P/LPV and a tumour-specific acquired somatic P/LPV in one of five hotspot codons on the wild type allele [55]. The germline P/LPV is inherited in an estimated 90% of cases, displaying an autosomal dominant inheritance pattern with reduced penetrance. *DICER1* encodes the “dicer” ribonuclease III, which is essential for the production of microRNAs and, therefore, plays a major role in regulating the expression of over 30% of protein-coding genes [56].

*DICER1* variants were initially implicated in the development of pleuropulmonary blastoma, but numerous other associated tumours and clinical features have been described since this discovery. These include macrocephaly, multinodular goitre (MNG), thyroid carcinomas (papillary, follicular, and poorly differentiated), tumours of the ovary and kidney, nasal and intestinal hamartoma, and pituitary and pineoblastoma [56].

By age 20 years, the cumulative incidence of MNG or thyroidectomy in *DICER1* carriers is 32% in females and 13% in males, rising to 75% and 17% by age 40. These individuals have a 16- to 24-fold risk of developing DTCs compared to the general population [57]. Among DTC patients, the frequency of *DICER1* P/LPVs appears to be higher in children than adults, having been only 0.6% in TCGA but 5–15% in paediatric studies [19,58,59]. These may occur as either compound heterozygous somatic variants or a germline and a somatic variant, and the identification of a somatic variant should trigger germline testing to determine the need for surveillance for other tumours.

*DICER1* P/LPVs are over-represented in follicular variant tumours. While FTCs account for only 10% of overall paediatric DTCs, in a study of 15 children with FTCs, 53% had a *DICER1* P/LPV, including all 3 children who were under the age of 10 years [60]. The diagnosis of an FTC in a young child should, therefore, prompt consideration of testing for *DICER1*.

Consensus guidelines recommend the consideration of testing for *DICER1* in any patient with a childhood onset MNG or DTC, especially if in combination with macrocephaly, another tumour typical of DICER1 syndrome, or a first-degree relative with an MNG/thyroid carcinoma or DICER1 syndrome features [55]. In asymptomatic individuals with DICER1 syndrome, thyroid ultrasound is recommended every 3 years from age 8 years, or more frequently if monitoring known nodules that appear to be low-risk. If undergoing chemo- or radiotherapy for another cancer, an ultrasound is recommended annually for 5 years from diagnosis [53]. FNA should be performed for identified nodules, as per paediatric guidelines.

### 3.3. Carney Complex Type 1

Carney complex type 1 (CC1) is a rare autosomal dominant genetic disorder caused by germline deactivating P/LPV in the tumour-suppressor gene *PRKAR1A*. This gene encodes the regulatory subunit of the cAMP-dependent protein kinase A (PKA), which has an important role in governing cell proliferation, differentiation, and apoptosis. Due to the deactivation of the regulatory subunit of PKA, the stimulation of endocrine tissues by their physiological stimulating hormones results in the unregulated activation of PKA, which leads to overactivity or neoplasia [61].

CC1 is characterised by mucocutaneous pigmentary anomalies; myxomas of the heart, skin, and breast; schwannomas; and various endocrine abnormalities, including primary pigmented nodular adrenocortical disease, large-cell calcifying Sertoli cell tumours, growth hormone-producing adenomas, and thyroid carcinomas or multiple adenomas [43]. Thyroid carcinomas are a relatively uncommon feature, occurring in ~3% of patients, and both PTCs and FTCs can be seen.

Thyroid surveillance with ultrasound every 1–2 years is recommended [62], although it is noted that ~75% of patients will have multiple nodules, most of which are non-functioning follicular adenomas [61]. Therefore, careful evaluation for suspicious features and the judicious use of FNA for suspicious nodules are needed.

### 3.4. Familial Adenomatous Polyposis

Familial adenomatous polyposis (FAP) is an autosomal dominant disorder caused by heterozygous P/LPV in *APC*, which interfere with the Wnt signalling pathway, disrupting several stages of cell development [63]. It is characterised by numerous colorectal adenomatous polyps, which inevitably progress to carcinomas, with or without extraintestinal manifestations. These include osteomas, dental abnormalities, dermoid tumours, adrenal masses, and tumours of the thyroid, liver, bile ducts, pancreas, and central nervous system [43].

The lifetime risk of thyroid carcinoma in FAP has mostly been reported as 1–2%, with a higher risk in females than males, although a 2007 study in 51 patients reported a prevalence of 12% in the context of ultrasound surveillance [64]. The mean age of diagnosis is 36–44 years. The pathognomonic form of FAP-associated thyroid carcinoma is cribriform-morular thyroid carcinoma (CMTC). Bilateral and multifocal carcinomas are frequent, but regional and distant metastases are uncommon [65]. Most FAP patients with thyroid carcinomas have a germline *APC* P/LPV in exon 15 [63]. Thyroid surveillance is recommended with ultrasound every 2 years from age 16 years [66].

### 3.5. Werner Syndrome

Werner syndrome is an autosomal recessive disorder of premature aging, including cataracts, grey hair, skin atrophy, and many diseases typically seen in old age occurring in middle age (30s), although, according to the diagnostic criteria, it can be diagnosed after age 10 [67]. It is caused by biallelic loss-of-function P/LPV in *WRN*, which has a role in DNA replication, transcription, recombination, and repair. Thyroid carcinomas are the most common neoplasms seen in Werner syndrome, along with melanoma, meningioma, sarcoma, primary bone tumours, and leukaemia/myelodysplasia [67].

The incidence of thyroid carcinomas in Werner syndrome is 8.9 times greater than the population risk, and there is a female predominance, with an average age of diagnosis of 39 years. FTCs are the most common variant; PTCs and ATCs occur as well [68].

### 3.6. Multiple Endocrine Neoplasia Type 2

Germline gain-of-function point mutations in *RET* are associated with multiple endocrine neoplasia type 2 (MEN2), while somatic mutations are found in sporadic MTCs, predominantly seen in adults. Mutations in the extracellular cysteine-rich region of the tyrosine kinase receptor, usually codons 609, 611, 618, 620, 630, or 634, cause MEN2A, an autosomal dominant, highly penetrant syndrome predisposing to early-onset MTC, bilateral phaeochromocytoma and hyperparathyroidism [16]. Several subtypes of MEN2A are described. MEN2A with cutaneous lichen amyloidosis is mostly caused by codon 634 mutations [69]. MEN2A with Hirschsprung disease can occur with exon 10 mutations; the mechanism leading to Hirschsprung disease is not clear [43]. Familial MTCs are associated with many MEN2A mutations but with lower penetrance, resulting in later-onset MTCs and the absence of phaeochromocytoma and hyperparathyroidism [69].

MEN2B is caused by the point mutation M918T or, in <5% of cases, A883F, and is associated with development of an MTC within the first year of life [69]. Individuals with MEN2B also have a high risk of phaeochromocytoma and present with a range of other features, including alacrima, constipation caused by intestinal ganglioneuromatosis, mucosal neuromas, and marfanoid facial features and body habitus [43]. While MEN2A mutations are usually inherited, 84% of M918T and 45% of A883F mutations are de novo [69].

MEN2 is rare; the prevalence of MEN2A is 13–24 per million and of MEN2B is 1–2 per million [69]. Sporadic MTCs are exceedingly rare in children, usually occurring after age 30, and the diagnosis of an MTC in a child or adolescent should prompt germline *RET* testing as they have MEN2 until proven otherwise [9]. Since MEN2B frequently occurs de novo, the recognition of the non-endocrine features is critical for diagnosis before a clinically apparent or metastatic MTC occurs [5,70].

The ATA currently divides *RET* point mutations into three risk categories for the purpose of guiding decisions about prophylactic thyroidectomy in individuals with inherited MEN2 [8]. M918T (MEN2B) carries the highest risk of an MTC, and total thyroidectomy in the first year of life is recommended [71]. Codon 634 and 883 mutations are considered high-risk, and thyroidectomy is recommended at age 5 years, or earlier if serum calcitonin is rising [8]. All other mutations are considered moderate-risk, and thyroidectomy is recommended when calcitonin is elevated with an upward trend, or earlier if the family do not wish to undertake long-term surveillance.

## 4. Germline Molecular Testing for Cancer Predisposition Syndromes

The diagnosis of CPS is important, as it allows for appropriate surveillance to be undertaken and the identification of family members at risk. The selection of paediatric patients with DTCs for germline testing is based on somatic profiling (the identification of a P/LPV in the tumour analysis suggestive of CPS); the recognition of syndromic features; a clinical presentation suggestive of CPS, including a multinodular goitre (PHTS and DICER1 syndrome); FTCs prior to 10 years of age (DICER1 syndrome); PDTCs (DICER1 syndrome); CMTCs (FAP); and/or a family history suggestive of a familial cancer predisposition. Macrocephaly is a highly relevant feature, being common in children with PHTS as well as DICER1 syndrome [53]. Therefore, head circumference should be measured routinely in children with paediatric thyroid carcinomas. A broad family history is essential to identify features of a CPS in family members, as their CPS may be undiagnosed. A panel of CPGs can be performed where a specific syndrome is not apparent clinically.

All children diagnosed with MTCs should be offered germline testing, as the majority of cases are associated with MEN2 syndrome.

## 5. Molecular Testing of Thyroid Nodule FNA Samples

Determining the likelihood of malignancy in a paediatric thyroid nodule with suspicious sonographic features traditionally relies on the cytological analysis of a sample obtained by fine needle aspiration (FNA). The results are categorised in regard to the risk of malignancy (ROM), utilising the Bethesda System for Reporting Thyroid Cytopathology, established in 2009. The 2023 update includes paediatric-specific ROM estimates for each of the six diagnostic categories (Table 1), which are higher within the indeterminate categories compared to the equivalent adult categories [72]. Approximately 35% of paediatric thyroid nodules undergoing FNA are categorised in one of the indeterminate categories: atypia of undetermined significance (AUS) or follicular neoplasm (FN) [6]. AUS includes the subcategories “nuclear atypia”, which is associated with higher ROM (59% in one study [73]), and “other”, which includes architectural, oncocytic, and lymphocytic atypia and is associated with a much lower ROM (as low as 6.5% [73]) [72].

The challenge is that the range of ROM in these two categories is very wide, 11–54% (mean 28%) in AUS and 28–100% (mean 50%) in FN [72]. Combining these factors with the sonographic risk features and patient risk factors can help to narrow down the estimated ROM, but there is no validated algorithmic method to do this in paediatric patients, and it presents a significant challenge to clinician decision-making. Most children in this situation have traditionally undergone a diagnostic lobectomy [6,72], although the 2022 European Thyroid Association paediatric guidelines included a recommendation to consider a repeat FNA in 6 months [2].

Further defining the risk of malignancy in paediatric thyroid nodules with indeterminate cytology, therefore, has significant potential to reduce unnecessary surgical procedures while maintaining a minimal risk of missed malignancy. The molecular testing of FNA samples with indeterminate cytology is now being integrated into standard practice for adult thyroid nodules [74] and has reduced the rates of diagnostic lobectomy by ~50–60% [75,76]. A similar proposal, utilising integrated data from ultrasound, cytology, and somatic oncogene alterations, has been proposed for paediatric patients [77].

The Afirma^®^ Gene Expression Classifier (Veracyte, Inc. South San Francisco, CA, USA) was released in 2011 and used the examination of messenger RNA and a proprietary algorithm to categorise patterns of gene expression with a “benign” or “suspicious” signature [74,78]. It was designed to be used as a “rule out” test to identify nodules that could safely be monitored without a diagnostic lobectomy, so had a high negative predictive value (NPV) of 94–95%. However, it had very low specificity, so many patients still underwent surgery and were ultimately diagnosed with a benign pathology. The second-generation platform, that includes the Afirma^®^ Xpression Atlas (oncogene and fusion panel to assess for an increased risk of malignancy) and the Gene Sequencing Classifier (GSC), uses next-generation sequencing with updated machine learning algorithms and maintains the excellent NPV while improving the PPV to 47% [79]. The GSC is not validated in patients < 21 years of age.

A DNA-based approach to molecular risk-stratification was also reported in 2011, using PCR and Sanger sequencing to examine for seven common point mutations and gene fusions, including *BRAF^V600E^* and *RAS* family point mutations, and *RET* and *PAX8::PPARG* fusions [80]. This approach was more useful to “rule in” malignancy and identify those patients who would benefit from an initial total thyroidectomy, given the strong association between these molecular drivers and malignancy, but those with a negative result still required a diagnostic lobectomy [74]. ThyroSeq^®^ (CBL Path, Inc. Rye Brook, NY, USA) used targeted next-generation sequencing (NGS) to expand this approach to include 284 mutational hot spots in 12 thyroid carcinoma genes [81]. ThyroSeq^®^ v2 added primers to detect mutations in *TERT*-p and RNA-sequencing with gene expression analysis of eight genes to quantify cell types and identify targeted gene fusions [74]. A further significant expansion of the platform resulted in ThyroSeq^®^ v3, which examines 112 genes for point mutations, insertions/deletions, gene fusions, copy number alterations, and gene expression and uses a genomic classifier to categorise nodules as benign or malignant [82]. This platform achieved an NPV of 97% and a PPV of 66% in a multicentre study of 247 Bethesda III and IV nodules [76].

Another commercially available multiplatform test is the combination of ThyGeNEXT^®^ and ThyraMIR^®^ (Interpace Diagnostics, Pittsburgh, PA, USA) [74,83]. The former is an expanded mutation panel test using targeted NGS to detect point mutations and fusions. If this returns a negative result, the ThyraMIR^®^ microRNA expression classifier test is performed to categorise the nodule as either low- or high-risk for malignancy. The combined test then reports a result of either negative, moderate, or positive. This test achieved an NPV of 95% and a PPV of 74% in a multicentre retrospective study of 178 Bethesda III and IV nodules [83]. ThyGeNEXT^®^ does not include analysis for *DICER1* or assess for novel fusion partners, and ThyraMIR^®^ has not been validated in paediatric patients.

Paediatric data on pre-surgical somatic molecular testing are limited, and none of the above platforms have been validated for use in children and adolescents. Given the differing molecular patterns seen in paediatric thyroid carcinomas and the higher pre-test probability of malignancy, these data cannot be directly extrapolated for use in children. The above studies were performed in nodules with a prevalence of malignancy of 24–30%. As paediatric indeterminate nodules have a higher ROM, the NPV is likely to be lower and, thus, they may not be suitable to rule out malignancy. On the other hand, the finding of a canonical P/LPV is clinically useful as the identification of a thyroid oncogene increases the likelihood of the nodule being a neoplasm or carcinoma. In addition, the risk of invasive, extrathyroidal disease does align with the P/LPV, allowing for the stratification of surgery [77].

Several retrospective paediatric studies have been published using the above testing platforms and showed insufficient sensitivity and NPVs to recommend their use as “rule out” tests [58,84,85]. However, they have improved over time, and Gallant et al., in 2022, were the first group to report that pre-operative testing using ThyroSeq^®^ v3 may allow clinicians to safely decrease diagnostic surgeries in children [11]. They reported a sensitivity of 96% and an NPV 95% after testing 95 formalin-fixed paraffin-embedded surgical specimens from patients < 22 years of age. Molecular testing missed two low-risk PTCs, giving a residual cancer risk of 4% in test-negative nodules, which they noted as being similar to the cancer risk in benign FNAs. It should be noted that they included patients > 18 years of age, and the median age was 16.3 years.

While there is a lack of prospective data to validate the use of pre-surgical somatic molecular testing in children and adolescents, many data are available to inform an evidence-based approach that can be implemented systematically. A recent proposal stratifies the risk of invasive behaviour in paediatric DTCs into three categories based on molecular alterations, rather than the two categories (*RAS*-like and *BRAF*-like) introduced by TCGA [77] (Table 2). This is based on the wealth of paediatric data that have been generated retrospectively, much of which is outlined in the previous sections. Tier 1, associated with a low risk of invasive behaviour, includes *RAS*, *DICER1*, *PTEN*, non-*V600E BRAF* point mutations, and *PAX8::PPARG* fusions; Tier 2, with an intermediate risk, includes *BRAF^V600E^* point mutations; Tier 3, with a high risk, includes *RET*, *NTRK*, *ALK*, and *BRAF* fusions. The point is made that the molecular findings are the only source of objective information about the risk of invasive behaviour, given the subjectivity of the reporting of ultrasound and cytology results.

In this proposal, nodules with indeterminate cytology and a Tier 1 molecular alteration and no sonographic evidence of lymph node metastasis would undergo lobectomy without prophylactic central neck dissection. Nodules with Tier 2 or 3 molecular alterations would undergo total thyroidectomy with prophylactic or therapeutic central neck dissection, but there may be consideration of lobectomy with prophylactic central neck dissection if there was no sonographic evidence of lymph node metastasis [77]. Management after the lobectomy would be dictated by the histological findings suggesting invasive behaviour or otherwise. Figure 1 illustrates this approach, noting that it covers the majority of paediatric thyroid nodules with Bethesda III or above cytology, but outliers with disparate cytology and molecular findings will occur, and their management should be guided by their cytology and then revised, if necessary, based on their histology. In patients with negative molecular testing on a comprehensive panel, surveillance should only be considered for nodules with Bethesda III cytology, due to limited data [11].

This approach is expected to reduce the extent of surgical intervention in patients with lower-risk disease, reducing their risk of complications and likely reducing costs to the health system. The choice of lobectomy, rather than total thyroidectomy, essentially removes the risk of hypoparathyroidism, and the need for lifelong levothyroxine therapy is unlikely. Where surgery can be avoided altogether, greater cost savings are expected. Pre-operative testing has been shown to be cost-effective in adults with indeterminate nodules, where it results in a substantial reduction in diagnostic lobectomies [86]. A formal cost analysis in the UK in 2022 found that pre-operative molecular testing became cost-effective at a price of GBP < 2177 [87], and this cut off is likely higher in children, given the potentially lifelong nature of surgical complications.

While this paradigm of routine pre-operative molecular testing is presented as an ideal approach to be adopted as widely as possible, it must be acknowledged that the cost and healthcare disparity across most health systems will preclude widespread access in the near future. Nevertheless, as the cost of molecular testing rapidly decreases and more data are available to demonstrate the impact of accurately stratifying a patient for surgery, this approach will likely see common adoption.

## 6. The Value of a Molecular Diagnosis Post-Operatively

Identifying somatic molecular alterations in confirmed thyroid carcinomas may serve several functions: to provide prognostic information about likely residual tumour behaviour and outcomes; to guide treatment decisions; and to enable the use of molecularly targeted therapies.

While there have been no prospective studies evaluating the utility of molecular analyses in paediatric thyroid carcinomas, there are relevant data from research into paediatric cancers more broadly. The first ZERO precision-guided treatment trial showed the significant benefit of molecular analysis in high-risk paediatric cancers (146 central nervous system [CNS] tumours, 183 solid tumours, and 56 haematological malignancies) [88]. Objective clinical benefit (OCB) was seen in 55% of children who received precision-guided treatment (PGT). Two-year progression-free survival (PFS) in those who received PGT was 26%, more than double that in those who received standard therapy (12%), and five times higher than that of those who received a novel agent not chosen on the basis of molecular findings (5.2%).

OCB and 2-year PFS were highest in those with a fusion or structural variant (75% and 68%, respectively), compared with single nucleotide variants (62% and 29%), high RNA target expression (46% and 5.9%), or copy number variation (29% and 7.7%). OCB and 2-year PFS were 56% and 18%, respectively, for solid tumours, compared with 43% and 27% for gliomas, 67% and 44% for other CNS tumours, and 40% and 0% for haematological malignancies. Importantly, the children who received their recommended therapy early in their treatment pathway (OCB 74%, 2-year PFS 42%) did significantly better than those who received it after their disease progressed (OCB 36%, 2-year PFS 12%). This study has now been extended to all children with cancer in Australia and New Zealand (ClinicalTrials.org ID NCT05504772).

The primary post-surgical utility of a molecular diagnosis is in those who are candidates for systemic medical therapy due to having advanced or RAI-resistant disease [2,74]. The identification of a *RET*, *NTRK*, *ALK*, or *MEK* fusion or a *BRAF^V600E^* point mutation enable the use of a molecularly targeted agent, while tumours without one of these canonical molecular alterations may be amenable to a multi-kinase inhibitor (MKI), although these agents are typically associated with more side-effects [89]. A minority of the clinical trials for molecular therapies published to date have included paediatric patients, and these data are supplemented by case reports.

Selpercatinib and pralsetinib inhibit the signalling of both mutant *RET* in MTCs and *RET* fusions in PTCs. Adult trials have shown high partial response rates and a few complete responses [90,91]. A recent clinical trial of selpercatinib included 10 paediatric patients with PTCs who showed an overall response rate (ORR) of 100% [92]. Paediatric case reports have shown efficacy to decrease tumour size [25], and pralsetinib has been used for the medical debulking of a large symptomatic MTC causing substantial airway and vessel narrowing [93]. It is FDA-approved for use from the age of 2 years for advanced or metastatic *RET*-mutant MTCs or *RET*-fusion positive RAI-refractory DTCs [94]. However, the FDA approval for pralsetinib in *RET*-mutant MTCs has been withdrawn as the phase III trial will no longer be pursued due to feasibility.

Larotrectinib is a TRK-inhibitor that has shown a partial response rate of 76%, with a 10% complete response rate and a 14% stable disease rate, in 21 patients with *NTRK*-fusion positive DTCs [95]. Two of these were aged <18 years, one of whom had a complete response and the other a partial response. It is FDA-approved for use from birth for locally advanced or metastatic *NTRK*-fusion positive solid tumours [96]. Entrectinib is a TRK, ALK, and ROS1-inhibitor that has shown a 54% ORR in adults with *NTRK*-fusion positive DTCs [97].

Dabrafenib decreases the activation of the MAPK pathway by binding to BRAF monomers in the setting of *BRAF^V600E^* mutations and is given in combination with trametinib, a MEK inhibitor, due to the limited durability of the response to BRAF-inhibitor monotherapy [98]. Clinical trials in adults with anaplastic thyroid carcinomas have shown an ORR of 56%, which is much higher than previous systemic therapy approaches [99]. No paediatric efficacy data are available to date.

The response rates to selective inhibitors are higher than those seen with MKIs, and they have been better tolerated, with very low rates of discontinuation due to side-effects. However, paediatric trial data are very limited. Table 3 shows a summary of FDA-approved agents for thyroid carcinomas with paediatric data, and Table 4 lists active clinical trials including children and adolescents. Several other agents are FDA-approved for paediatric use in the absence of paediatric trial data [89].

Recent data show that <20% of paediatric DTC patients with pulmonary metastasis treated with RAI achieve complete remission, and chronic, stable residual disease is common [23,103,104,105]. In addition to their direct effect on disease activity, molecularly targeted therapies can stimulate the redifferentiation of thyroid carcinoma cells. There are multiple case reports of molecularly targeted therapies successfully inducing the re-expression of the sodium iodide symporter in RAI-refractory PTCs, thus increasing the RAI-uptake and, possibly, the efficacy of RAI therapy [25,106,107,108,109,110]. Clinical trials are underway to assess whether the incorporation of molecularly targeted therapies prior to RAI can enhance the efficacy of RAI in paediatric patients with pulmonary metastasis and improve the percent of children that achieve a complete response (ClinicalTrials.gov IDs NCT05024929 and NCT05783323).

With this in mind, in addition to considering adjuvant targeted therapy for patients with structurally progressive, RAI-refractory PTCs that are not amenable to surgery (mostly distant metastases), we should continue to investigate the potential of using targeted therapies in the neoadjuvant setting, as they have successfully been used to reduce tumour burden in large, inoperable tumours and enabled successful surgery [93]. There is much work and prospective research still to be done to establish the indications, duration of treatment, and durability of response for these new and powerful therapeutic agents, which will be made possible as molecular testing for paediatric thyroid carcinoma patients becomes more widespread.

## 7. Conclusions and Future Directions

Our understanding of the molecular basis of paediatric thyroid carcinoma is rapidly evolving. The body of data now available supports the integration of somatic molecular testing into the risk-stratification of thyroid nodules and the management of advanced disease.

The identification of a molecular alteration associated with a low risk of invasive disease in a thyroid nodule with indeterminate cytology supports a less extensive surgical approach, with a reduced risk of complications and cost. Nodules with a molecular alteration associated with an intermediate or a high risk of invasive disease require a more extensive surgical approach, informed by the thorough sonographic assessment of cervical lymph nodes.

The identification of CPSs is important to allow surveillance for patients at risk of further cancers and identify family members at risk, and it informs management decisions for patients with known thyroid nodules and carcinomas. A detailed family history and knowledge of CPSs are necessary to inform the selection of candidates for germline testing.

The development of molecularly targeted therapies is introducing a new era of personalised, precision medicine for children and adolescents with thyroid carcinomas, especially those with advanced and metastatic disease. These agents have the potential to improve outcomes by restoring RAI-uptake in RAI-refractory disease and reducing the tumour burden in the adjuvant and neoadjuvant settings. Prospective research is needed and is underway to further define and confirm the role of molecular testing and molecularly targeted therapies in paediatric thyroid carcinomas.

As the paradigms for the assessment and management of paediatric thyroid nodules and carcinomas become more nuanced and personalised, multidisciplinary collaboration is critical to support clinical care and research.

## Figures and Tables

**Figure 1 cancers-17-01549-f001:**
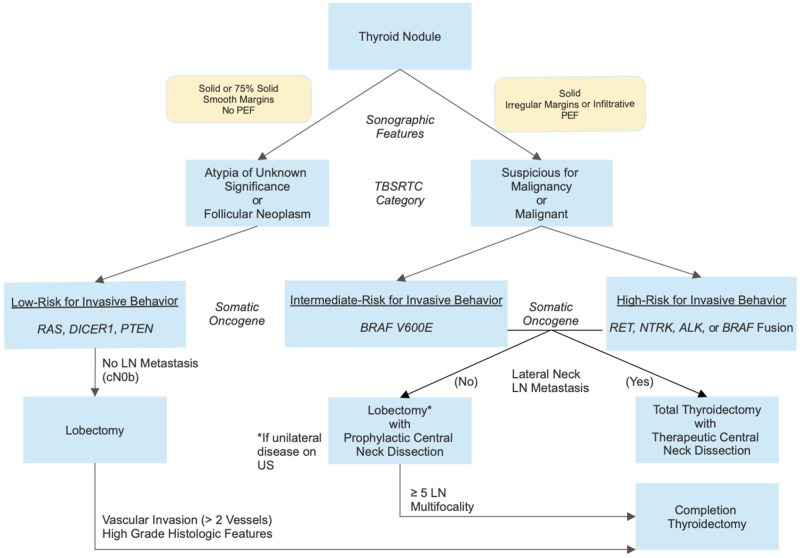
The incorporation of pre-operative molecular data across the 3-tiered paediatric risk of invasive behaviour of differentiated thyroid carcinomas to stratify the surgical management of a thyroid nodule. High-grade histologic features include solid, trabecular, or insular growth patterns; a mitotic index of ≥ 3 per 10 high power fields; necrosis; and convoluted nuclei. Reprinted with permission from Ref. [77]. 2025, Oxford University Press. PEF: punctate echogenic foci; TBSRTC: The Bethesda System for Reporting Thyroid Cytopathology; US: ultrasound.

**Table 1 cancers-17-01549-t001:** 2023 Bethesda system for reporting thyroid cytopathology in paediatric patients with implied risk of malignancy (ROM).

Diagnostic Category		ROM Mean (Range)
I	Nondiagnostic	14% (0–33)
II	Benign ^a^	6% (0–27)
III	Atypia of undetermined significance	28% (11–54)
IV	Follicular neoplasm ^b^	50% (28–100)
V	Suspicious for malignancy	81% (40–100)
VI	Malignant	98% (86–100)

Adapted with permission from Ref. [72]. 2023, Mary Ann Liebert, Inc. ^a^ ROM is skewed by selection bias because most thyroid nodules classified as benign do not undergo surgical excision. ^b^ Includes cases of follicular neoplasm with oncocytic features (formerly Hurthle cell neoplasm).

**Table 2 cancers-17-01549-t002:** Risk of invasive disease and suggested surgical approach based on molecular alterations in paediatric DTCs.

Tier	Molecular Alteration	Risk of Invasive Disease	Surgical Approach	
1	Point mutations *RAS* *DICER1* *PTEN* *BRAF* (non-*V600E*) Fusions *PAX8::PPARG*	Low	Lobectomy without central neck dissection provided no evidence of LN metastasis on US	Completion thyroidectomy if >2 vessel vascular invasion or high-grade histologic features
2	Point mutations *BRAF^V600E^*	Intermediate	Total thyroidectomy with prophylactic/therapeutic central neck dissection. Consider lobectomy with prophylactic central neck dissection if no evidence of LN metastasis on US.	Completion thyroidectomy if multifocality or ≥5 positive LNs
3	Fusions *RET* *NTRK* *ALK* *BRAF*	High		

High-grade histologic features include solid, trabecular, or insular growth patterns; a mitotic index of ≥3 per 10 high power fields; necrosis; and convoluted nuclei. Post-surgical risk-stratification, management, and follow-up as per American Thyroid Association Paediatric Guidelines. LN: lymph node; US: ultrasound.

**Table 3 cancers-17-01549-t003:** Paediatric clinical trial data for molecularly targeted agents.

Agent	Molecular Target(s)	Histology	n	Age (Years)	Response	Citation
Selpercatinib	RET	MTC	14	2–20 2–20	ORR 83.3%	[92]
		PTC	10		ORR 100%	
					2-year PFS 92.4%	
Larotrectinib	NTRK	Agnostic	78	0.1–17.8	ORR 88%	[100]
			94	0–18	ORR 84%	[101]
		PTC	2	6–13	PR 50%, CR 50%	[95]
Cabozantinib	VEGFR2, RET, MET, FLT3, NTRK, AXL	MTC	5	1	PR 40%	[102]

CR: complete response; ORR: objective response rate; PFS: progression-free survival; PR: partial response.

**Table 4 cancers-17-01549-t004:** Current trials of molecularly targeted therapies for paediatric thyroid carcinomas.

Agent	Molecular Target(s)	Histology	Age (Years)	Location(s)	Status	ClinicalTrials.gov ID
Larotrectinib	NTRK	DTC	≥1	USA	Recruiting	NCT05783323
Repotrectinib	ALK, ROS1, NTRK	Agnostic	≥12	International	Recruiting	NCT03093116
Pralsetinib	RET	Agnostic	0.5–21	International	Active, not recruiting	NCT03899792
Pralsetinib	RET	Thyroid carcinoma	≥12	USA	Active, not recruiting	NCT04759911
Pralsetinib	RET	DTC	≥12	USA	Recruiting	NCT05668962
Pralsetinib	RET	Agnostic	≥12	International	Active, not recruiting	NCT03157128
Oncogene-specific kinase inhibitors	NTRK, RET, ALK, *BRAF^V600E^*	DTC	≥0	USA, Australia	Recruiting	NCT05024929

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
