# Peer review of "Paediatric Thyroid Carcinoma: The Genetic Revolution and Its Implications for Therapy and Outcomes"

_cancers, 2025, doi:10.3390/cancers17091549_

Round 1

Reviewer 1 Report

Comments and Suggestions for Authors

Authors provide a review on the genetic landscape of paediatric tyroid carcinomas. Authors describe several somatic gene alterations,  germline P/LPV associated with TC. Then they describe  some syndromes association thyroid cancer. Then they describe molecular testing options. both pre- and post operatively.

 The review is interesting but need to be re organized and some clarifications are more than welcome.

For example  at page 10 it is suggested a classification of thyroid oncogenes in three categories with  low, intermediate and high  risk of invasive behavior. It woulf be  better that Table 2 undelines this classificaiton proposal with adapted surgical approach and follow-up recommandations according to the risk. Table 2 to be re-organized.

Regarding the value of molecular diagnosis post-operatively, table 3 cites TKI agents and studeis in adult population which is not adapted for pediatric population.

Instead of this table authors should further detail i the ZERO trial as well as provide a table related to studies of molecular targets in pediatric population describing the age of treated paediatric population and the percentage of objective response  as well as PFS.

A table with ongoing research trials realted to targeted agents for thyroid cancer in pediatric population would be also suitable.

In conclusion authors should detail the advantages ans limits of molecular testing in pediatric DTC.

Author Response

Dear editors,

Re: Response to reviewers, manuscript ID: cancers-3535898

Thank you for inviting us to resubmit this revised manuscript following peer review. We are grateful for the reviewers’ feedback and comments, and provide our responses below.

Reviewer 1

Authors provide a review on the genetic landscape of paediatric thyroid carcinomas (TC). Authors describe several somatic gene alterations, germline P/LPV associated with TC. Then they describe some syndromes association thyroid cancer. Then they describe molecular testing options. both pre- and post-operatively.

Comment 1

The review is interesting but need to be re organized and some clarifications are more than welcome. For example, at page 10 it is suggested a classification of thyroid oncogenes in three categories with low, intermediate and high risk of invasive behavior. It would be better that Table 2 underlines this classification proposal with adapted surgical approach and follow-up recommendations according to the risk. Table 2 to be re-organized.

Response 1

Thank you for this recommendation. Table 2 has been re-written to reflect the proposed risk-stratification and surgical approach based on molecular alterations. Post-surgical management and follow-up should still be guided by ATA paediatric guidelines.

Comment 2

Regarding the value of molecular diagnosis post-operatively, table 3 cites TKI agents and studies in adult population which is not adapted for pediatric population. Instead of this table authors should further detail in the ZERO trial as well as provide a table related to studies of molecular targets in pediatric population describing the age of treated paediatric population and the percentage of objective response as well as PFS.

Response 2

Further detail on ZERO trial has been added, including tumour types included in the trial and further detail on outcomes (Page 20, paragraphs 4 & 5). Table 3 has been amended to focus on paediatric trial data, which are admittedly limited.

Comment 3

A table with ongoing research trials related to targeted agents for thyroid cancer in pediatric population would be also suitable.

Response 3

Thank you. This has been added as Table 4.

Comment 4

In conclusion authors should detail the advantages and limits of molecular testing in pediatric DTC.

Response 4

We have re-written and re-organised the final section and especially the conclusion to more clearly illustrate and detail these aspects.

Reviewer 2 Report

Comments and Suggestions for Authors

The analyse can be specialzed to gene analyses but based on the genetic analyses of selected cojortes without the cohese to large cancer pediatric  TC

thw disscussion used any old and not so relevant refernces on the second hand the new knowlidges from the past 5 years and more relevant were ignored. The last but not least fact are rhat the text is very closed to the work published in the Diagnostc and Endocrine relevant cancer journal and or the Thyroid (USA). 
that the knowledges in this papers are not in literature but the real senteces are similer. Nothing new infromation only the modficate senteces. No original idea not novelity nothig reelvant to the ATA recommandation 2015 that the BRAf is recommand to the de-escaltion but this paper come not with some new paradogmas to rhat. 

Author Response

Reviewer 2

Comment 1

The analyse can be specialzed to gene analyses but based on the genetic analyses of selected cojortes without the cohese to large cancer pediatric TC thw disscussion used any old and not so relevant refernces on the second hand the new knowlidges from the past 5 years and more relevant were ignored.

Response 1

We thank the reviewer for the comments and would like to point out that 60% of references cited in the manuscript were published within the last 5 years. However, if there are specific references the reviewer believes should be cited we would be delighted to review them and accordingly include them. Nonetheless, we believe this article is a fair representation of the current landscape in paediatric thyroid cancer.

Comment 2

The last but not least fact are that the text is very closed to the work published in the Diagnostic and Endocrine relevant cancer journal and or the Thyroid (USA).

Response 2

Thank you. As we have endeavoured to present a harmonised view of the current landscape in paediatric thyroid cancer, it is not surprising that the information we present resonates with that of other leading publications. The senior authors are from a diverse range of disciplines and across continents, and we have endeavoured to include data from as many continents as possible, including Asia, Europe and North and South America.

Comment 3

that the knowledges in this papers are not in literature but the real senteces are similer. Nothing new infromation only the modficate senteces. No original idea not novelity nothig reelvant to the ATA recommandation 2015 that the BRAf is recommand to the de-escaltion but this paper come not with some new paradogmas to rhat.

Response 3

Thank you. It is worth pointing out that the majority of this article focuses on the impact of molecular genetics and our improved understanding of tumour drivers in the diagnosis, risk-stratification, phenotype and treatment of paediatric thyroid cancers. This paradigm of personalised medicine and risk-stratification is totally novel compared with that of the 2015 guidelines, which are in great need of an update, and that, we are aware, is imminent. Furthermore, it is irrefutable that paediatric tumour drivers and phenotypes are markedly different to those in adult thyroid cancers, which we also highlight in this article.

Editor requests

  1. Please provide Simple Summary.

This has been added prior to the abstract.

  1. Please confirm that all the images and tables are not reprinted/ have not
    been previously published. If they were, please make sure that permission has
    been obtained and there is no copyright issue.

Table 1 and Figure 1 are reprinted and written permission has been obtained for this. Legends have been amended to reflect this.

  1. Please add "Conclusions" section.

The final section has been renamed so that it is clearly labelled as the conclusion.

  1. Please add Funding information to the back matter of the manuscript.

This has been added following author contributions.

Once again, thank you for the opportunity to resubmit and we look forward to your response.

Round 2

Reviewer 1 Report

Comments and Suggestions for Authors

Authors have improved theri mansucript

Reviewer 2 Report

Comments and Suggestions for Authors

No comment